# Multimodal Cancer Survival Analysis with Learnable Queries

## Abstract

Leveraging multimodal data, particularly the integration of whole-slide histology images (WSIs) and transcriptomic profiles, holds great promise for improving cancer survival prediction. However, excessive redundancy in multimodal data poses a critical challenge for model optimization and can become prohibitive. Thus, methods that effectively reduce redundancy are highly desirable. While previous approaches have achieved impressive results by clustering redundant representations, they still rely on additional prior knowledge, which limits their flexibility in capturing dynamic data changes and emerging patterns. To resolve this drawback, we propose a novel and effective approach, SurvQ, for multimodal cancer survival analysis with learnable queries, which adaptively learns representative features in a data-driven manner, reducing redundancy while preserving critical information. Our method employs two sets of learnable query vectors that serve as a bridge between high-dimensional representations and survival prediction, capturing task-relevant features. Additionally, we introduce a multimodal mixed self-attention mechanism to enable cross-modal interactions, further enhancing information fusion. Extensive experiments on five benchmark cancer datasets demonstrate that our method consistently outperforms state-of-the-art approaches, achieving the best average performance.

## 1 Introduction

Survival analysis, a cornerstone of patient prognostic modeling, aims to predict the time until an event of interest occurs (typically death), thereby improving therapeutic decision-making, optimizing patient care, and aiding in the identification of novel biomarkers associated with disease progression (Song et al., 2023). Prognostication is a complex challenge influenced by diverse perspectives (Chen et al., 2022b). Multimodal methods that integrate features from histology and genomics data can offer complementary insights, capturing subtle changes that may remain undetected within single-modality analyses (Zhou & Chen, 2023; Chen et al., 2022b; 2021; Xu & Chen, 2023; Jaume et al., 2024; Zhang et al., 2024; Song et al., 2024; Zhou et al., 2025). 0.686). 0.686). Histology provides detailed phenotypic insights into cell types and the tumor microenvironment (Wang et al., 2021b; Li et al., 2022b). Genomics data, such as bulk transcriptomics (Acosta et al., 2022), represents gene expression, revealing a comprehensive landscape of molecular information (Chen et al., 2022b; Lipkova et al., 2022; Steyaert et al., 2023; Song et al., 2024). Since these modalities capture distinct aspects of tumor biology, their integration enables a more holistic characterization of disease progression.

In recent years, various multimodal methods (Chen et al., 2021; 2022b; Zhou & Chen, 2023; Xu & Chen, 2023; Jaume et al., 2024) have combined these two modalities to enhance precision in risk stratification and optimize survival analysis (Figure 1(a)). However, these works are hampered by the extensive number of histology and genomic tokens (*e.g.*, patches of WSIs and pathways of gene expression), leading to information redundancy issue (Hosseini et al., 2024; Zhang et al., 2024). Several approaches (Song et al., 2024; Zhang et al., 2024; Zhou et al., 2025) have been proposed to incorporate additional knowledge by clustering tokens into fixed categories, thereby reducing the number of cross-modal tokens (Figure 1(b)). We refer to these representative categories as **prototype-based** methods. For instance, MMP (Song et al., 2024) groups all histology tokens into morphology-related categories, while PIBD (Zhang et al., 2024) and CCL (Zhou et al., 2025)

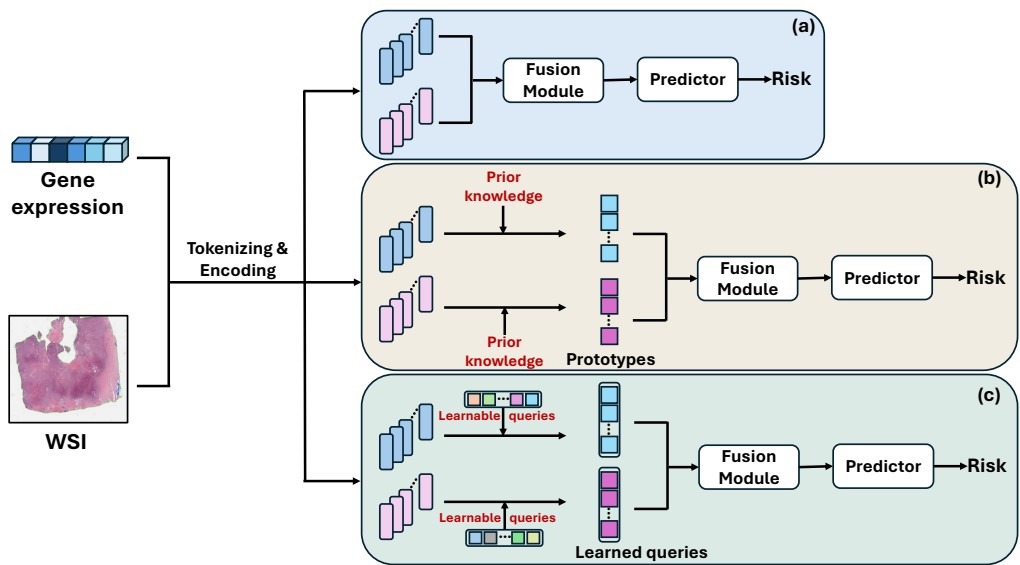

Figure 1: Illustration of typical multimodal cancer survival analysis architectures: (a) Directly fusing multimodal data through a fusion module, such as an attention mechanism (*e.g.*, SurvPath (Jaume et al., 2024)). (b) Reducing redundant tokens from cross-modal data using additional knowledge, such as predefined risk levels (*e.g.*, PIBD (Zhang et al., 2024)). (c) Our proposed approach adaptively learns task-relevant information with learnable queries.

cluster large token sets based on risk levels and censorship knowledge. Although these methods significantly reduce the number of cross-modal tokens, they remain suboptimal in compacting extensive histology and genomic information. This limitation arises from their reliance on predefined prototypes based on morphology, risk levels, or censorship, restricting their flexibility in capturing dynamic data changes and emerging patterns.

To address these limitations, we propose **SurvQ**, a straightforward yet effective approach that learns representative features in a data-driven manner (Figure 1(c)). Our method begins by extracting uni-modal representations of pathology and genomics data, following (Jaume et al., 2024). To mitigate redundancy without relying on additional knowledge, we introduce two sets of learnable query vectors that interact with pathology and genomic features through cross-attention mechanisms. The pathology and genomic queries extract compact representations, enabling the model to distill essential features from high-dimensional data. The learnable queries act as a bridge between high-dimensional representations and survival analysis, capturing task-relevant features while minimizing redundancy without requiring additional knowledge. To further enhance multimodal fusion, we employ a multimodal mixed self-attention mechanism on the combined set of histology and genomic queries, enabling the model to learn cross-modal interactions and improve information fusion.

We summarize the contributions as follows: (1) We propose SurvQ, a novel multimodal framework designed to mitigate information redundancy in cancer survival analysis. (2) To achieve this, we introduce two sets of learnable queries for extracting pathology and genomic representative features. We leverage cross-attention to capture task-relevant information and multimodal mixed self-attention mechanism to model cross-modal interactions. (3) Extensive evaluations on five benchmark cancer datasets demonstrate the effectiveness of SurvQ, achieving the best average performance.

## 2 RELATED WORK

### 2.1 SINGLE-MODALITY SURVIVAL ANALYSIS

Survival analysis is critical for patient prognostication, with the goal of learning risk estimates that accurately rank survival times. Whole-slide images (WSIs) and genomics data each provide essential and complementary information for this task. In the early stages, many attempts focused on exploiting single-modality data to model survival outcomes (Zhu et al., 2017; Wulczyn et al., 2020; Yao

et al., 2020a; Yousefi et al., 2017; Qiu et al., 2020). For example, (Zhu et al., 2017) firstly develop an end-to-end way to predict survival based on WSIs. (Yao et al., 2020a) proposed a siamese MI-FCN combined with attention-based MIL pooling to extract informative features from WSIs and aggregate them for patient-level survival prediction. (Yousefi et al., 2017) integrated deep learning with Bayesian optimization to predict cancer outcomes from high-dimensional genomic data. (Qiu et al., 2020) proposed a meta-learning framework for survival analysis with limited high-dimensional samples. While single-modality methods achieve strong performance in survival prediction, they may overlook crucial cross-modal information.

## 2.2 Multi-modality survival analysis

Recent works (Mobadersany et al., 2018; Wang et al., 2021a; Li et al., 2022a; Chen et al., 2022b; 2021; Xu & Chen, 2023; Jaume et al., 2024; Zhang et al., 2024; Song et al., 2024; Zhou et al., 2025) demonstrate that multimodal approaches can yield superior predictive performance, robustness, and a more holistic characterization of cancer prognosis. Existing fusion strategies are generally classified as either tensor-based or attention-based methods (Zhang et al., 2020; 2024). Tensor-based fusion methods, including concatenation (Mobadersany et al., 2018), weighted summation (Huang et al., 2020), bilinear pooling (i.e., Kronecker product)(Wang et al., 2021a), and factorized bilinear pooling(Li et al., 2022a), are typically applied at either early or late fusion stages, which often fail to fully capture inter-modal interactions (Jaume et al., 2024; Song et al., 2024; Zhang et al., 2024). Attention-based fusion methods have recently been developed to capture cross-modal correlations through co-attention mechanisms (Chen et al., 2021; Jaume et al., 2024; Song et al., 2024; Zhang et al., 2024). For example,(Chen et al., 2021) employed a cross-attention module to learn genomics-related features from WSIs, but it only modeled patch-to-gene interactions and suffered from significant gene overlap across sets. To better represent dense interactions between modalities,(Jaume et al., 2024) introduced biological pathways to tokenize genes, thereby reducing the number of gene tokens. However, this approach remained constrained by the large number of histology tokens. To address this limitation,(Song et al., 2024) aggregated histology patches and biological pathways into prototypes defined by morphology and cancer hallmarks. Similarly,(Zhang et al., 2024; Zhou et al., 2025) clustered large token sets into prototypes guided by risk levels and censorship knowledge. While these **prototype-based** methods effectively reduced the token number for cross-attention, their reliance on prior knowledge such as morphology, risk levels, or censorship restricts their flexibility in modeling dynamic data changes and emerging patterns.

## 2.3 Multimodal learning with learnable queries

In recent years, the concept of learnable queries, a fixed set of trainable embeddings that interact with modality-specific features through cross-attention, has become a powerful mechanism for learning compact representations. At the beginning, DETR (Carion et al., 2020) introduced learnable queries as a data-driven alternative to the massive anchor-based pre-setting mechanism for object detection. Later, BLIP-2 (Li et al., 2023) adopted a similar mechanism in its Q-Former to compress redundant tokens from multimodal data (e.g., image and text) for vision–language learning. In this context, learnable queries serve as a compact information bottleneck that adaptively extracts task-relevant features while reducing redundancy. QnA (Arar et al., 2022) uses queries solely as local attention compressors to reduce spatial redundancy in images. In multimodal medical learning, learnable queries have been introduced as an effective way to bridge heterogeneous data (Peng et al., 2025; Li et al., 2025; van Sonsbeek et al., 2023; Wei et al., 2024). NEARL-CLIP (Peng et al., 2025) employs cross-modality queries that interact with both image and text embeddings, enabling the model to align visual cues with clinical semantics. Similarly, MedBridge (Li et al., 2025) integrates learnable query tokens to selectively extract and align complementary information across medical images and associated clinical reports. In survival analysis, G-HANet (Wang et al., 2024) employs learnable queries to capture histo-genomic associations by reconstructing genomic signals during training. At inference time, these learned queries act as surrogates for genomic information, enabling the model to operate without requiring genomic inputs.

Intuitively, learnable queries provide a simple yet effective solution for reducing redundancy. Hence, we introduce a **query-based** approach that, to the best of our knowledge, is the first to employ learnable queries for token reduction in multimodal cancer survival analysis, where whole-slide images (WSIs) and genomic data often produce extremely high-dimensional representations. Inspired by

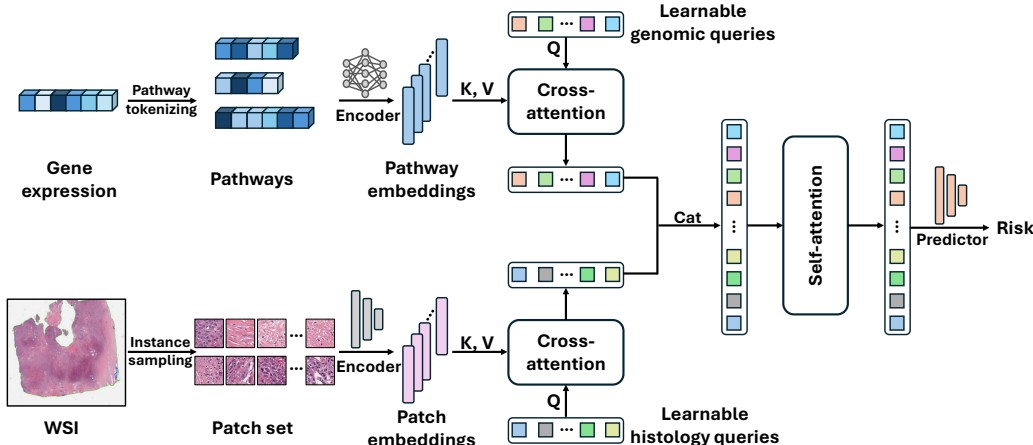

Figure 2: Overview of **SurvQ**. Gene expression is first tokenized into biological pathways, and pathway embeddings are extracted using a feature extractor (SNN (Klambauer et al., 2017)). Similarly, WSIs are processed into patch embeddings using a pre-trained feature extractor. Next, an adaptive prototyping module employs two sets of learnable queries to extract compact information from high-dimensional representations via cross-attention. These learned queries are then fused using a multimodal mixed self-attention mechanism, facilitating cross-modal interactions and enhancing information integration. Finally, the model predicts survival risk based on the refined queries.

DETR (Carion et al., 2020) and BLIP-2 (Li et al., 2023), our SurvQ introduces learnable queries that adaptively capture the most task-relevant information from both WSIs and genomic data while minimizing redundancy, all without relying on prior knowledge.

## 3 METHODS

We introduce the **SurvQ** framework, which employs learnable queries to adaptively extract informative representations from histology and genomics data for survival prediction. First, we describe adaptive prototyping mechanism with learnable queries in Section 3.1. Then, we outline the multimodal fusion mechanism and describe survival prediction in Section 3.3. Figure 2 shows the framework of our SurvQ.

### 3.1 UNIMODAL FEATURE EXTRACTION

We start by extracting unimodal features from histology and genomic data to reduce the information redundancy. The details are outlined below.

**Histology.** WSIs capture detailed tissue phenotypes, offering critical insights for cancer prognosis prediction. At $20\times$ magnification, a single WSI can reach resolutions up to $150{,}000 \times 150{,}000$ pixels (Chen et al., 2022a). To process such large images, we first identify tissue regions and exclude background areas that lack diagnostic relevance. The remaining tissue is divided into a set of $N_H$ non-overlapping patches, $H = \{h_1, \ldots, h_{N_H}\}, \quad N_H > 10^4$. Directly storing and processing all patches is impractical, so we extract patch embeddings before training. Specifically, we use the pre-trained UNI model (Chen et al., 2024) as a feature extractor $f(\cdot)$ to map each patch $h_i$ into an embedding $x_i^{(H)} = f(h_i)$. A learnable linear projection is then applied to obtain the final patch token $X^{(H)} \in \mathbb{R}^{N_H \times D}$. Collectively, this process provides a compact yet semantically meaningful representation of WSIs for downstream survival prediction.

**Genomics.** Bulk transcriptomics captures gene expression patterns that reflect the molecular state of a tumor, including its aggressiveness and response to treatment. These molecular signatures provide valuable prognostic information, making transcriptomics a powerful tool for predicting patient survival. Given a set of gene-level transcriptomics $T = \{t_1, \ldots, t_{N_T}\}$, where $N_T$ is the number of

measured genes, we follow (Jaume et al., 2024) and group genes with known functional interactions into pathways, $P = \{p_1, \ldots, p_{N_P}\}$, where $N_P$ denotes the number of pathways. To encode pathways of variable length into fixed-size representations, we employ self-normalizing neural networks (SNNs) (Klambauer et al., 2017) as feature extractors $g_i(\cdot), i \in \{1, \ldots, N_P\}$. Each pathway $p_i$ is encoded by $x_i^{(G)} = g_i(t_{p_i})$, where $t_{p_i}$ denotes the set of gene features in pathway $p_i$. Collectively, this yields the final pathway token $X^{(G)} \in \mathbb{R}^{N_P \times D}$, where each pathway token encodes its constituent genes into a biologically interpretable and end-to-end learnable representation for survival prediction.

## 3.2 Learnable histology and genomics queries.

While patch and pathway tokens provide valuable information for survival analysis, their sheer volume limits the effective use of attention mechanisms for capturing comprehensive patterns. To address this, we introduce *learnable queries* that adaptively summarize high-dimensional histology and genomics features into compact, task-relevant representations.

For histology, we initiate a set of learnable queries $Q^{(H)} \in \mathbb{R}^{N_{Q_h} \times D}$ that interact with patch embeddings $X^{(H)}$ through a cross-attention mechanism. Here, $Q^{(H)}$ serve as queries, while $X^{(H)}$ act as both keys and values. The resulting compact histology representation is obtained as

$$Q'^{(H)} = \text{Softmax}\left(\frac{F_q^{(H)}(Q^{(H)}) F_k^{(H)}(X^{(H)})^\top}{\sqrt{D}}\right) F_v^{(H)}(X^{(H)}), \tag{1}$$

where $F_q^{(H)}, F_k^{(H)}, F_v^{(H)}$ are linear projections for queries, keys, and values in the histology branch.

Similarly, for genomics we initiate a set of learnable queries $Q^{(G)} \in \mathbb{R}^{N_{Q_g} \times D}$ that interact with pathway embeddings $X^{(G)}$. The compact genomic representation is given by

$$Q'^{(G)} = \text{Softmax}\left(\frac{F_q^{(G)}(Q^{(G)}) F_k^{(G)}(X^{(G)})^\top}{\sqrt{D}}\right) F_v^{(G)}(X^{(G)}), \tag{2}$$

where $F_q^{(G)}, F_k^{(G)}, F_v^{(G)}$ are linear projections for the genomic branch.

Intuitively, the learnable queries $Q'^{(H)}$ and $Q'^{(G)}$ act as adaptive information bottlenecks: instead of processing thousands of patch and pathway tokens directly, they distill the most prognostically relevant tissue patterns from WSIs and molecular signals from transcriptomics into compact feature sets that can be more effectively fused for survival prediction.

## 3.3 Multimodal fusion

To capture dense interactions between the compact histology and genomic representations, we introduce a multimodal mixed self-attention mechanism. Specifically, we form a multimodal sequence by concatenating the compact features from both modalities,

$$M^{(HG)} = Q'^{(H)} \| Q'^{(G)} \in \mathbb{R}^{(N_{Q_h} + N_{Q_g}) \times D}$$

where $N_{Q_h} + N_{Q_g} \ll N_H + N_P$. Thus, the sequence length after query aggregation is greatly reduced compared to the original patch and pathway tokens, enabling efficient and effective multimodal attention. Applying the standard self-attention formulation (Vaswani et al., 2017), we obtain the fused sequence

$$M'^{(HG)} = \text{Softmax}\left(\frac{F_q^{(HG)}(M^{(HG)}) F_k^{(HG)}(M^{(HG)})^\top}{\sqrt{D}}\right) F_v^{(HG)}(M^{(HG)}), \tag{3}$$

where $F_q^{(HG)}, F_k^{(HG)}, F_v^{(HG)}$ are projection functions for the multimodal branch.

Unlike prior multimodal transformers that decompose attention into separate intra- and cross-modal components (Jaume et al., 2024), our approach directly applies self-attention over the concatenated representations. This allows all histology and genomic features to interact within a unified attention space, promoting dense cross-modal fusion while avoiding the overhead of multiple attention modules. Finally, we average the fused sequence to obtain the final multimodal representation $z^{(HG)} \in \mathbb{R}^{(N_{Q_h} + N_{Q_g})}$.

### 3.4 SURVIVAL PREDICTION

Given the fused multimodal representation $z^{(HG)}$, we aim to predict patient survival following the standard discrete-time survival formulation (Zadeh & Schmid, 2021). The survival state of each patient is characterized by two elements: (1) the event indicator $c$, where $c = 0$ denotes an observed death and $c = 1$ denotes censoring at the last follow-up, and (2) the time-to-event $t$, which corresponds to the time from diagnosis to death when $c = 0$, or to the last follow-up when $c = 1$. Rather than predicting the continuous time-to-event $t$ directly, we discretize the timeline into $n$ non-overlapping intervals $[t_{k-1}, t_k)$, $k = 1, \ldots, n$, based on the quartiles of observed survival times, and denote the corresponding interval label as $y_k$. The survival prediction task is thus reformulated as a classification problem with censoring. Each patient sample is represented as $(H, y_k, c)$, where $H = \{h_1, \ldots, h_n\}$ denotes the predicted hazard vector, with each element $h_j$ giving the probability that the event occurs in the corresponding interval $[t_{j-1}, t_j)$. Additionally, we define the discrete survival function as $f_{\text{surv}}(H, y_k) = \prod_{j=1}^{k}(1 - h_j)$.

Following (Jaume et al., 2024; Zhang et al., 2024), we define the negative log-likelihood (NLL) survival loss (Zadeh & Schmid, 2021) as

$$\mathcal{L}_{\text{surv}} = -c \log\left(f_{\text{surv}}(H, y_k)\right) - (1 - c)\log\left(f_{\text{surv}}(H, y_k - 1)\right) - (1 - c)\log\left(h_{y_k}\right).$$

## 4 EXPERIMENTS

### 4.1 DATASETS.

We performed extensive experiments using five public cancer datasets from The Cancer Genome Atlas (TCGA)[1]: Breast Invasive Carcinoma (BRCA, n=869), Bladder Urothelial Carcinoma (BLCA, n=359), Head and Neck Squamous Cell Carcinoma (HNSC, n=392), Colon and Rectum Adenocarcinoma (COADREAD, n=296), and Stomach Adenocarcinoma (STAD, n=317). We trained models to predict disease-specific survival (DSS) (Jaume et al., 2024), which more accurately reflects the patient's disease status compared to overall survival. For histology data, we extracted non-overlapping $224 \times 224$ patches at 20× magnification. For genomic data, raw transcriptomics were obtained from the Xena database (Goldman et al., 2020), along with DSS labels. 331 human biological pathways were collected, represented as transcriptomics sets with specific molecular interactions, sourced from the Human Molecular Signatures Database (MSigDB) - Hallmarks (Liberzon et al., 2015; Subramanian et al., 2005) (50 pathways from 4,241 genes) and Reactome (Gillespie et al., 2021) (281 pathways from 1577 genes), ensuring at least 90% of transcriptomics were accessible.

### 4.2 EVALUATION METRICS.

To reduce potential batch artifacts, we use 5-fold cross-validation for each dataset. Model performance is evaluated using the concordance index (C-index) (Harrell et al., 1996) and its standard deviation (std), which measures the accuracy of ranking patients based on their survival months and predicted risk. In addition, Kaplan-Meier (Kaplan & Meier, 1958) analysis is applied to examine survival outcomes and assess differences between the predicted high- and low-risk groups.

### 4.3 IMPLEMENTATION DETAILS.

The proposed algorithm is implemented in Python with Pytorch library and runs on a single NVIDIA A100 GPU. UNI (Chen et al., 2024), a DINOv2-based ViT-Large (Oquab et al., 2024) model pre-trained on $1 \times 10^8$ patches sampled from $1 \times 10^5$ WSIs collected at Mass General Brigham, is used as the feature extractor to get 1024-dimensional embeddings. We further use an MLP with a 512-dimensional hidden layer as the latent vector encoder to embed patch features into a fixed dimension of 256. Meanwhile, the feature extractors of pathways are SNNs following the settings in works (Jaume et al., 2024; Song et al., 2024; Zhang et al., 2024; Zhou et al., 2025). All models are trained with a $5 \times 10^{-4}$ learning rate with $1 \times 10^{-3}$ weight decay for 50 epochs, AdamW op-

---

[1]https://portal.gdc.cancer.gov/

Table 1: Comparison of SurvQ and baseline methods for disease-specific patient survival prediction, measured by the C-Index. The best performance is highlighted in bold. * Indicates prototype-based methods.

| Model | BRCA | BLCA | COADREAD | HNSC | STAD | Avg. |
|---|---|---|---|---|---|---|
| *Genomic* | | | | | | |
| MLP | $0.598 \pm 0.063$ | $0.501 \pm 0.071$ | $0.709 \pm 0.158$ | $0.512 \pm 0.057$ | $0.479 \pm 0.052$ | 0.560 |
| SNN | $0.639 \pm 0.067$ | $0.584 \pm 0.067$ | $0.732 \pm 0.134$ | $0.567 \pm 0.055$ | $0.557 \pm 0.051$ | 0.616 |
| *Histology* | | | | | | |
| AMISL | $0.613 \pm 0.046$ | $0.601 \pm 0.053$ | $0.694 \pm 0.123$ | $0.602 \pm 0.054$ | $0.559 \pm 0.032$ | 0.614 |
| ABMIL | $0.642 \pm 0.065$ | $0.612 \pm 0.065$ | $0.702 \pm 0.148$ | $0.619 \pm 0.048$ | $0.608 \pm 0.054$ | 0.636 |
| *Multimodal* | | | | | | |
| Porpoise | $0.642 \pm 0.043$ | $0.619 \pm 0.056$ | $0.702 \pm 0.143$ | $0.631 \pm 0.042$ | $0.639 \pm 0.075$ | 0.646 |
| MCAT | $0.713 \pm 0.033$ | $0.632 \pm 0.066$ | $0.715 \pm 0.158$ | $0.635 \pm 0.098$ | $0.668 \pm 0.087$ | 0.673 |
| MOTCat | $0.712 \pm 0.042$ | $0.641 \pm 0.067$ | $0.728 \pm 0.134$ | $0.641 \pm 0.064$ | $0.658 \pm 0.066$ | 0.676 |
| SurvPath | $0.723 \pm 0.045$ | $0.642 \pm 0.054$ | $0.726 \pm 0.161$ | $0.646 \pm 0.057$ | $0.649 \pm 0.051$ | 0.677 |
| PIBD* | $0.716 \pm 0.026$ | $0.650 \pm 0.067$ | $0.734 \pm 0.153$ | $0.642 \pm 0.054$ | $0.656 \pm 0.051$ | 0.680 |
| MMP* | $0.746 \pm 0.064$ | $0.660 \pm 0.050$ | $0.741 \pm 0.168$ | $0.641 \pm 0.046$ | $0.640 \pm 0.037$ | 0.686 |
| CCL* | $0.772 \pm 0.066$ | $0.662 \pm 0.055$ | $0.758 \pm 0.118$ | $0.629 \pm 0.047$ | $0.632 \pm 0.053$ | 0.690 |
| HSFSurv | $0.771 \pm 0.061$ | $0.672 \pm 0.031$ | $0.761 \pm 0.125$ | $0.651 \pm 0.042$ | $0.674 \pm 0.058$ | 0.705 |
| MoME | $0.768 \pm 0.063$ | $0.666 \pm 0.022$ | $0.785 \pm 0.124$ | $0.640 \pm 0.054$ | $0.673 \pm 0.057$ | 0.706 |
| **SurvQ** | $\mathbf{0.794 \pm 0.062}$ | $\mathbf{0.677 \pm 0.060}$ | $\mathbf{0.812 \pm 0.115}$ | $\mathbf{0.653 \pm 0.045}$ | $\mathbf{0.686 \pm 0.053}$ | **0.724** |

timizer (Kingma & Ba, 2017) and the batch size is set to 32. we set the number of histology and genomic learnable queries to 300 and 128.

### 4.4 QUANTITATIVE EVALUATION.

**C-index comparison:** We evaluate our method against three groups of SOTA approaches: (1) Unimodal Methods: For genomic data, we use MLP (Haykin, 1998) and SNN (Klambauer et al., 2017) as baselines. For histology, we compare against ABMIL (Ilse et al., 2018) and AMISL (Yao et al., 2020b). (2) Multimodal Methods: We benchmark our approach against six leading multimodal models: Porpoise (Chen et al., 2022b), MCAT (Chen et al., 2021), MOTCat (Xu & Chen, 2023), SurvPath (Jaume et al., 2024), MoME (Xiong et al., 2024) and HSFSurv (Fu et al., 2026). (3) Prototype-based Methods: We further compare our method with three prototype-based multimodal approaches: PIBD (Zhang et al., 2024), MMP (Song et al., 2024), and CCL (Zhou et al., 2025).

The results, summarized in Table 1, indicate that SurvQ consistently outperforms all other methods across five cancer datasets, achieving the highest average C-index of 0.724. Compared to unimodal approaches, SurvQ surpasses the best-performing genomic and histology models (SNN: 0.616, AB-MIL: 0.636), respectively, highlighting the advantage of integrating multimodal information and the significance of effectively mitigating information redundancy.

Among multimodal methods, SurvQ achieves the highest performance across all six benchmarks, surpassing the second-best method (MoME: 0.706) by 1.8 percentage points in average C-index. Furthermore, compared with the prototype-based multimodal group, SurvQ demonstrates clear superiority, achieving performance gains ranging from 1.5 to 5.4 percentage points compared to CCL. By leveraging learnable queries as an intermediary between high-dimensional representations, SurvQ effectively mitigates information redundancy and enhances survival prediction, confirming its effectiveness in multimodal cancer analysis.

**Kaplan-Meier analysis:** We further assess the prognostic ability of our model through Kaplan-Meier (KM) survival analysis, as shown in Figure 3. Patients are stratified into high- and low-risk groups according to their predicted risk scores, with the median score of each validation set used as the cutoff. The resulting KM curves reveal clear separation between the two groups across all five TCGA datasets, with low-risk patients consistently exhibiting longer survival times. The statistical significance of these differences is confirmed by log-rank tests, with all p-values below 0.05.

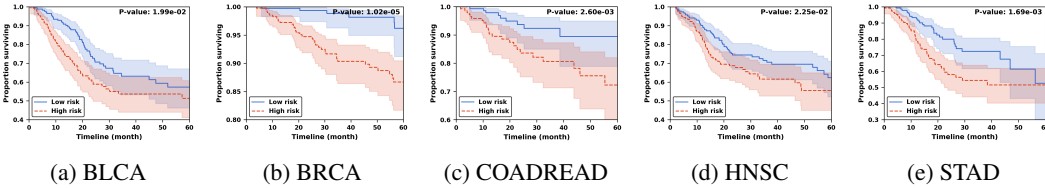

(a) BLCA      (b) BRCA      (c) COADREAD      (d) HNSC      (e) STAD

Figure 3: Kaplan–Meier curves comparing predicted high-risk (red) and low-risk (blue) groups. Shaded bands denote confidence intervals; $p < 0.05$ indicates statistical significance.

Notably, the BRCA, COADREAD, and HNSC datasets show particularly strong group separation, underscoring the robustness of our approach in discriminating survival outcomes. These results highlight the effectiveness of our model in stratifying patients into clinically meaningful subgroups.

## 4.5 ABLATION STUDY.

We conduct a series of ablation studies on BRCA, BLCA, and COADREAD datasets to evaluate the impact of three key components of SurvQ: learned histology queries, learned genomics queries, and the multimodal mixed self-attention mechanism.

As illustrated in Tabel 2, we begin with a simple baseline that directly concatenates the extracted histology and genomic features, followed by a predictor for survival prediction. This baseline achieves an average performance of 0.709. Introducing learnable histology queries (Hist.) reduces redundancy in histology representations and improves the average score to 0.734, highlighting their effectiveness. Adding learnable genomic queries (Geno.) provides further gains, with consistent improvements across all datasets and an average score of 0.745. Finally, incorporating the multimodal mixed self-attention mechanism (Self-attn.) enables iterative cross-modal interactions and more effective fusion, yielding the best overall performance of 0.761. Additional ablation studies are provided in the Appendix.

Table 2: Ablation study on learnable queries of SurvQ.

| Hist. | Geno. | Self-attn. | BRCA | BLCA | COADREAD | Avg. |
|:---:|:---:|:---:|:---:|:---:|:---:|:---:|
| | | | $0.724 \pm 0.0612$ | $0.651 \pm 0.034$ | $0.754 \pm 0.146$ | 0.709 |
| ✓ | | | $0.745 \pm 0.0787$ | $0.660 \pm 0.051$ | $0.779 \pm 0.144$ | 0.734 |
| ✓ | ✓ | | $0.761 \pm 0.0624$ | $0.673 \pm 0.032$ | $0.801 \pm 0.1426$ | 0.745 |
| ✓ | ✓ | ✓ | $\mathbf{0.794 \pm 0.062}$ | $\mathbf{0.677 \pm 0.060}$ | $\mathbf{0.812 \pm 0.115}$ | $\mathbf{0.761}$ |

## 4.6 MODEL BEHAVIOR VISUALIZATION.

To gain an intuitive understanding of SurvQ, we analyze its behavior by examining cross-attention maps and learned queries for both histology and genomics on representative TCGA-STAD and TCGA-HNSC cases, as shown in Figure 4.

For histology (Panels B and E), we visualize the attention maps of two randomly selected learned queries, where patch embeddings serve as keys and values. The brighter regions on the whole-slide images indicate higher query attention, revealing how different queries specialize in distinct tissue regions. Importantly, the corresponding representative patches (bottom of each panel) highlight that different queries capture diverse morphological patterns such as glandular structures, stromal textures, and cellular density, demonstrating the model's ability to disentangle heterogeneous histological features within the same tumor.

For genomics (Panels C and F), we examine the top six pathways associated with two randomly chosen learned genomic queries. Each query is linked to distinct biological processes, reflecting how the model organizes high-dimensional transcriptomic signals into interpretable functional modules. For instance, in STAD (Panel C), genomic queries are enriched for pathways involved in energy metabolism (e.g., ATP formation, proton cotransport), DNA repair, and cellular trafficking, while in HNSC (Panel F), they capture hallmark oncogenic processes such as TGF-beta signaling, DNA

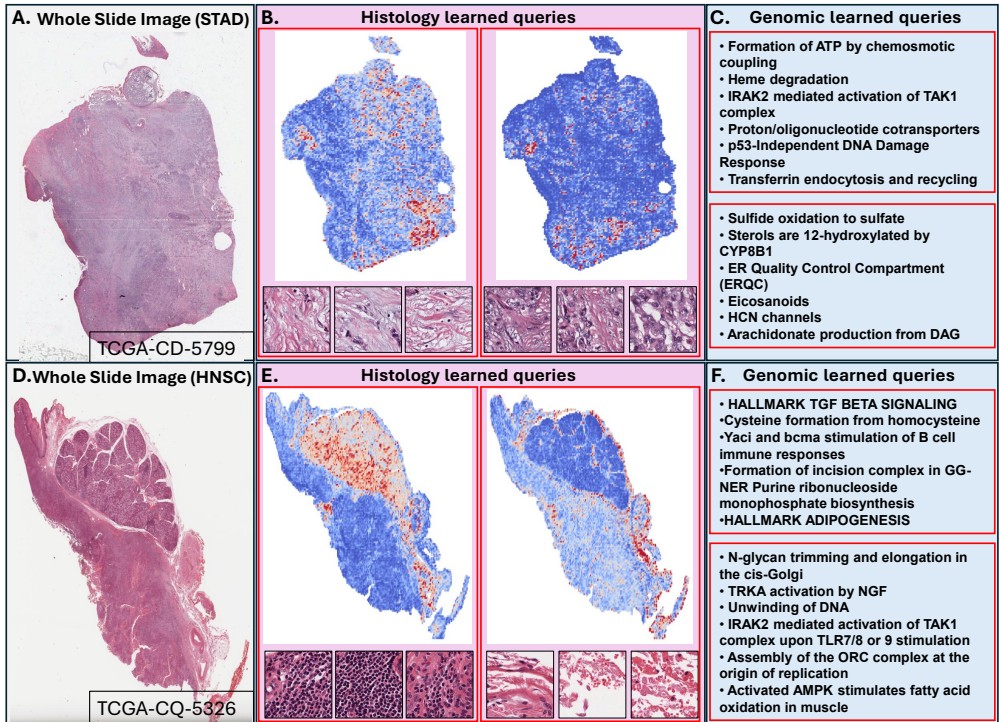

Figure 4: Visualization of SurvQ's learned queries for histology and genomics. (A, D) Whole-slide images (WSIs) from representative TCGA-STAD and TCGA-HNSC patients. (B, E) Cross-attention maps of two randomly selected histology queries, where higher-intensity (warmer-color) regions indicate stronger relevance to the query. The bottom panels show the three most representative patches for each query, highlighting distinct morphological patterns. (C, F) The top six pathways associated with two randomly selected genomic queries, illustrating that different queries capture diverse and biologically meaningful processes. Histology and genomic queries are highlighted with red boxes.

damage repair, and immune response activation. This suggests that learned genomic queries not only reduce redundancy but also align with biologically meaningful processes that are central to cancer progression.

Together, these visualizations highlight that SurvQ learns complementary and interpretable representations: histology queries localize diverse morphological phenotypes, while genomic queries uncover key molecular pathways. This dual interpretability provides confidence that the model is not merely fitting survival labels but is also capturing biologically relevant tumor characteristics across modalities.

## 5 CONCLUSION AND LIMITATIONS

In this paper, we introduced SurvQ, a novel approach for multimodal cancer survival analysis that reduces redundancy while preserving critical information. It employs learnable query vectors to capture task-relevant features and a multimodal mixed self-attention mechanism to enhance cross-modal interactions. Experiments on five benchmark cancer datasets confirm the superiority of SurvQ over existing methods, highlighting its effectiveness in improving cancer survival prediction.

While our method reduces the number of tokens, it requires a fixed number of learnable queries for both histology and genomics data across all datasets. This constraint may not be optimal, and exploring the dynamic number of queries remains an avenue for future work.

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

# A  APPENDIX

We disclose our use of LLMs in Section A.1, provide additional ablations on learnable queries and inference efficiency in Section A.2, and present more model-behavior visualizations in Section A.3.

## A.1  DISCLOSURE OF LLM USAGE

We used large language models for grammar refinement and revision assistance. All conceptual contributions, experimental design, data analyses, and conclusions are entirely the work of the authors.

## A.2  MORE ABLATION STUDY

**Learnable query numbers:**  Another pivotal aspect of SurvQ is the number of learnable queries. To assess its effect, we vary the query numbers for both modalities and evaluate performance. As shown in Table 3, the model benefits from a sufficient pool, about 300 queries for histology and 128 for genomics, while larger numbers offer diminishing returns. Notably, all configurations still surpass prior methods.

Table 3: Ablation study on different numbers of learnable query of SurvQ

| Hist. | Geno. | BRCA | BLCA | COADREAD | HNSC | STAD | Avg. |
|---|---|---|---|---|---|---|---|
| 400 | 128 | $0.743 \pm 0.046$ | $0.657 \pm 0.037$ | $0.813 \pm 0.095$ | $0.669 \pm 0.042$ | $0.655 \pm 0.032$ | 0.708 |
| 300 | 128 | $0.794 \pm 0.062$ | $0.677 \pm 0.047$ | $0.812 \pm 0.079$ | $0.653 \pm 0.066$ | $0.686 \pm 0.051$ | 0.724 |
| 200 | 128 | $0.792 \pm 0.045$ | $0.652 \pm 0.053$ | $0.780 \pm 0.099$ | $0.655 \pm 0.050$ | $0.666 \pm 0.025$ | 0.709 |
| 300 | 256 | $0.751 \pm 0.082$ | $0.673 \pm 0.057$ | $0.764 \pm 0.119$ | $0.677 \pm 0.051$ | $0.687 \pm 0.036$ | 0.710 |
| 300 | 128 | $0.794 \pm 0.062$ | $0.677 \pm 0.047$ | $0.812 \pm 0.079$ | $0.653 \pm 0.066$ | $0.686 \pm 0.051$ | 0.724 |
| 300 | 64 | $0.800 \pm 0.039$ | $0.679 \pm 0.045$ | $0.807 \pm 0.098$ | $0.649 \pm 0.052$ | $0.646 \pm 0.010$ | 0.716 |

**Learnability of queries:**  We evaluate the effectiveness of learnable queries by introducing a Fixed Query baseline, where the queries are randomly initialized and kept frozen (non-learnable). As shown in Table 4, the learned queries consistently achieve higher performance. This demonstrates that learnable queries adapt to dataset-specific multimodal structures and enable more effective information aggregation than a static, non-adaptive query set.

Table 4: Ablation study on fixed vs. learnable queries.

| | BRCA | COADREAD |
|---|---|---|
| Fixed query | $0.768 \pm 0.057$ | $0.778 \pm 0.140$ |
| **Ours** (Learnable query) | $\mathbf{0.794 \pm 0.062}$ | $\mathbf{0.812 \pm 0.115}$ |

**Inference Efficiency.**  We present an inference efficiency analysis in Table 5. All inference times are measured on a single NVIDIA A100 GPU. Compared with PIBD, SurvQ achieves substantially faster inference (106 ms vs. 172 ms on BRCA; 106 ms vs. 170 ms on COADREAD) and lower GPU memory usage (1.21 GB vs. 1.42 GB). This highlights the advantage of our simple formulation and lightweight architecture: compressing multimodal features into a compact set of queries not only preserves predictive performance but also delivers meaningful improvements in computational efficiency.

Table 5: Ablation study on average inference time (ms), GPU memory (GB).

| | BRCA | | COADREAD | |
|---|---|---|---|---|
| | Inference time ($\downarrow$) | GPU ($\downarrow$) | Inference time ($\downarrow$) | GPU ($\downarrow$) |
| PIBD | 172 | 1.42 | 170 | 1.42 |
| SurvQ | **106** | **1.21** | **106** | **1.21** |

Table 6: PCA-based information preservation analysis on COADREAD dataset.

| Modality | Var(Original features) | Var(Learned queries) |
|---|---|---|
| Histology | 99.05% | 94.55% |
| Omics | 99.04% | 96.70% |

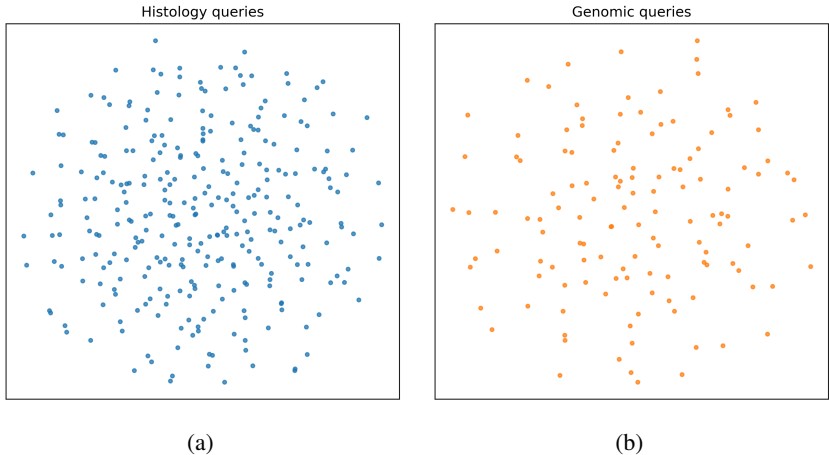

(a)  (b)

Figure 5: t-SNE visualization of the learned queries. The pathology queries (a) and genomic queries (b) form well-separated and diverse clusters, indicating that the model learns distinct modality-specific representations rather than collapsing into a shared embedding space.

**Information preservation analysis.** To assess how much information is preserved by the learnable queries, we perform PCA on the original histology and omics features and compute the variance explained by their principal subspace. We then project the learned queries onto this PCA basis and measure the proportion of variance retained within the shared subspace. As shown in Table 6, the queries retain 94.6% of the histology variance and 96.7% of the omics variance captured by the original representations. The difference between 99.05% and 94.55% reflects the fact that the raw patch features contain substantial high-frequency appearance variations (e.g., staining variability, morphological noise, tile-level artifacts) that contribute to the overall variance but are largely irrelevant to survival prediction. The learned queries deliberately suppress these noisy or task-irrelevant directions during end-to-end learning, focusing instead on the lower-dimensional, task-relevant subspace. For omics features, the variance gap between the original representation (99.04%) and the learned queries (96.7%) is smaller than in histology. This is expected because omics embeddings are already low-noise, highly structured, and free from spatial or staining artifacts. These results demonstrate that SurvQ achieves effective information-preserving compression: the learned queries substantially reduce the token count while maintaining nearly all major modes of variation present in the original features.

### A.3 MORE VISUALIZATION ON MODEL BEHAVIOR

**Interpretability of queries.** To assess whether the learned queries capture meaningful structure from the multimodal data, we visualize their geometry using t-SNE (van der Maaten & Hinton, 2008). Figure 5 shows a 2D embedding of 300 histology and 128 genomic learned queries.

The two modalities form clearly separated yet internally diverse clusters, indicating that the model automatically organizes the queries into modality-specific subspaces that capture distinct histological and genomic characteristics. The broad dispersion within each modality further suggests that the learned queries encode heterogeneous, non-redundant patterns rather than collapsing into a few dominant directions. Taken together, these results show that the queries form a diverse and discriminative representation space, supporting the interpretability of SurvQ and its ability to meaningfully factorize multimodal information.

**More examples of unimodal query.** Similar to Figure 4, Figure 6 provides additional examples of SurvQ's learned queries for histology and genomics on BRCA, BLCA, and COADREAD.

**Model behavior of cross-modal interactions.** We further analyze the behavior of **SurvQ**'s cross-modal interactions between histology and genomics to better understand and interpret how SurvQ learns multimodal relationships. To illustrate these interactions, we randomly select one learnable genomic query and list its associated genomic pathways. We then use the multimodal mixed self-attention to identify the most related histology queries and map them back to the original image patches to visualize the most representative regions.

As shown in Fig. 7 and 8, the learned genomic semantics (B) align well with the biologically coherent histology queries (C, D). For instance, in the BLCA case, the cross-modal associations identified by SurvQ reveal biologically interpretable correspondences between genomic pathways and histological patterns. Pathways related to lipid and steroid metabolism are predominantly linked to necrotic or lipid-rich tumor regions, which typically exhibit pale, loosely structured tissue morphology. In contrast, the NODAL/TGF-$\beta$ signaling pathway corresponds to epithelial-stromal transition zones, reflecting regions where tumor cells interact with surrounding stromal components and where epithelial-mesenchymal-transition (EMT)-related biology is expected to occur. Additionally, pathways associated with purine salvage and telomere synthesis map to highly proliferative tumor areas characterized by densely packed, hyperchromatic nuclei.

These observations collectively demonstrate that the learned cross-modal queries in SurvQ capture meaningful, biologically grounded interactions between histology and genomics, going beyond what standard unimodal models can explain.

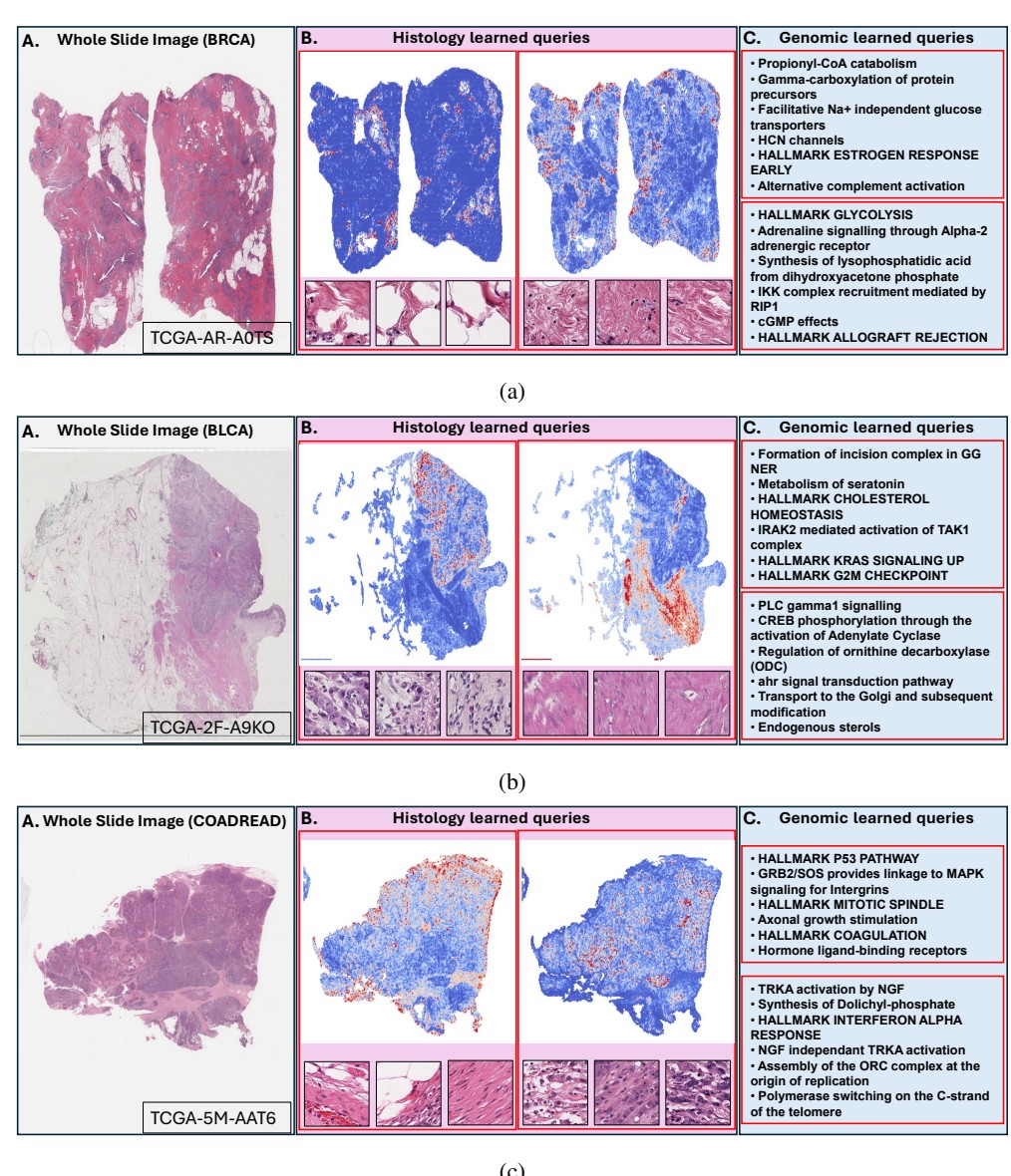

Figure 6: Visualization of SurvQ's learned queries for histology and genomics.

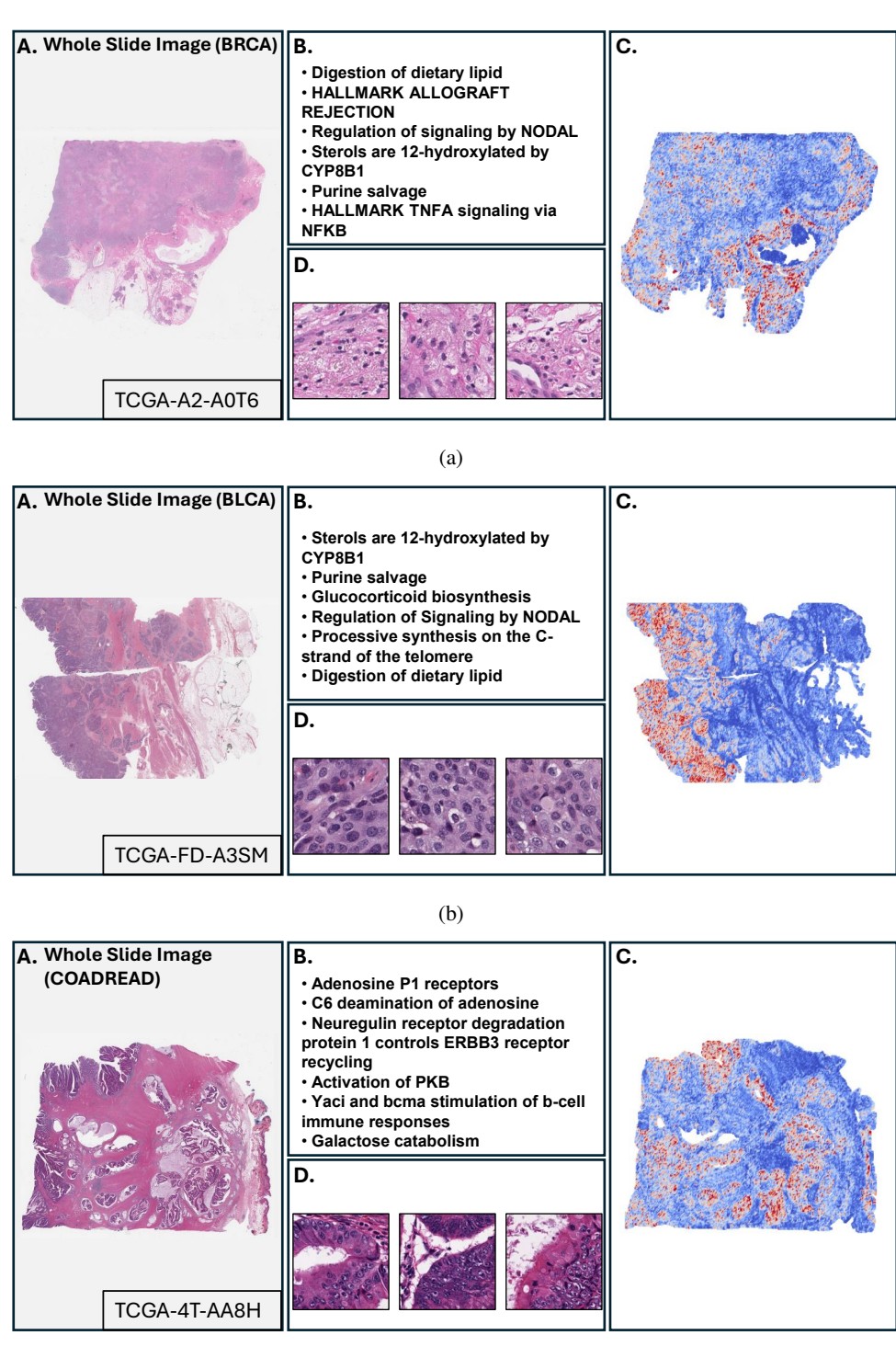

Figure 7: Cross-modal interactions learned by SurvQ across multiple TCGA datasets.

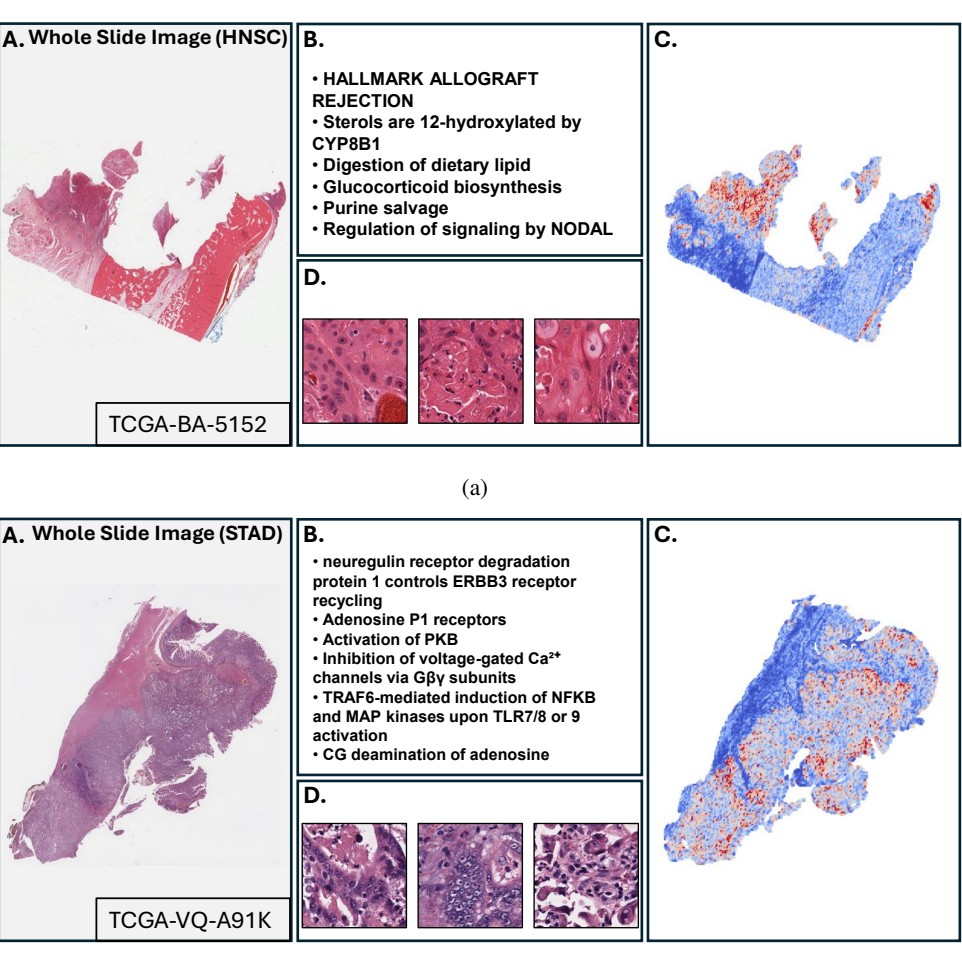

Figure 8: Cross-modal interactions learned by SurvQ across multiple TCGA datasets. For each dataset, we visualize one representative case to illustrate how the model discovers biologically meaningful correspondences between genomic queries and histology patterns after self-attention fusion. (A) Whole-slide image (WSI) of the selected case. (B) Top pathways associated with the genomic query that receives the highest aggregated attention from pathology queries in the self-attention matrix. (C) Spatial heatmap indicating which pathology patches are most strongly attended by this genomic query after multimodal fusion. (D) Representative high-attention patches extracted from the highlighted regions in (C). Across all datasets, SurvQ consistently learns coherent cross-modal relationships, with specific genomic queries attending to biologically plausible histology regions, demonstrating both the interpretability and generality of the query-based multimodal interaction mechanism.

