# OpenReview forum: "Multimodal Cancer Survival Analysis with Learnable Queries"
_ICLR.cc/2026/Conference — Submitted to ICLR 2026_

### Official Review · Reviewer_ubYy · 2025-10-15

**Soundness:** 3
**Presentation:** 2
**Contribution:** 2
**Rating:** 2
**Confidence:** 3

**Summary:**

This paper introduces SurvQ, a novel framework for multimodal cancer survival analysis that addresses the critical challenge of information redundancy in histology (WSI) and genomic data. Unlike previous methods that rely on predefined, knowledge-based prototypes to cluster data, SurvQ employs a data-driven approach using "learnable queries." The model utilizes two sets of learnable query vectors—one for histology and one for genomics—that interact with high-dimensional patch and pathway tokens via cross-attention. This mechanism acts as an adaptive information bottleneck, distilling vast and redundant inputs into a compact, task-relevant set of representative features. These learned queries are then fused using a unified multimodal mixed self-attention module to capture complex cross-modal interactions efficiently. Experiments conducted on five benchmark TCGA cancer datasets show that SurvQ achieves superior predictive performance.

**Strengths:**

1. Architectural Elegance and Efficiency: The proposed architecture is both effective and relatively simple. By using queries to drastically reduce the number of tokens before fusion, it allows for the use of a single, powerful "multimodal mixed self-attention" mechanism.

2. Superior Empirical Performance: The method's effectiveness is strongly supported by the results.

3. Good Interpretability: The paper provides clear visualizations demonstrating that the learnable queries capture biologically meaningful information.

**Weaknesses:**

1. Fixed Number of Queries: A limitation, acknowledged by the authors, is that the number of learnable queries for each modality is a fixed hyperparameter.
2. Reducing the redundancy of the attention mechanism through learnable queries is common in other fields[1]. I think this design is not novel enough.
[1] Arar M, Shamir A, Bermano A H. Learned queries for efficient local attention[C]//Proceedings of the IEEE/CVF Conference on Computer Vision and Pattern Recognition. 2022: 10841-10852
3. There is no ablation experiment on the number of queries
4. The article highlights the redundancy in the input but does not provide a detailed analysis of training time or computational resource usage compared to other methods (e.g. flops), which makes it difficult to quantify whether the redundancy is addressed.
5. There are relatively few baselines for comparison, and most are not from 25 years ago.

**Questions:**

1. Fixed Number of Queries: A limitation, acknowledged by the authors, is that the number of learnable queries for each modality is a fixed hyperparameter.
2. Reducing the redundancy of the attention mechanism through learnable queries is common in other fields[1]. I think this design is not novel enough.
[1] Arar M, Shamir A, Bermano A H. Learned queries for efficient local attention[C]//Proceedings of the IEEE/CVF Conference on Computer Vision and Pattern Recognition. 2022: 10841-10852
3. There is no ablation experiment on the number of queries
4. The article highlights the redundancy in the input but does not provide a detailed analysis of training time or computational resource usage compared to other methods (e.g. flops), which makes it difficult to quantify whether the redundancy is addressed.
5. There are relatively few baselines for comparison, and most are not from 25 years ago.

---

> ### Author Response · Authors · 2025-11-24
> **Response to Reviewer ubYy**
>
> We thank the reviewer for the insightful comments.
>
> **Q1: Fixed Number of Queries: A limitation, acknowledged by the authors, is that the number of learnable queries for each modality is a fixed hyperparameter.**
>
> A1: We follow the common practice in DETR-style architectures by using a fixed number of queries across all datasets to ensure fair comparison and reproducibility. Furthermore, as shown in Supplementary Table 3, we conducted an ablation study on the number of queries. Although the optimal choice varies slightly across datasets, our selected configuration provides consistently strong performance and offers a good balance between accuracy and stability. For these reasons, we adopt a unified setting for all datasets. We will explore adaptive query numbers in future work.
>
> **Q2: Reducing the redundancy of the attention mechanism through learnable queries is common in other fields. I think this design is not novel enough.**
>
> A2:  Thank you for the comment. We fully agree that reducing redundancy through learnable queries has been explored in other domains, including DETR, BLIP-2, Mask2Former, and SAM. These methods all employ learnable queries to extract compact task-specific representation. For example, object-centric slots in DETR, prompt-aligned regions in SAM, or locally compressed tokens in QnA.
>
> However, the role, motivation, and function of learnable queries in SurvQ differ substantially from these prior works. QnA, for instance, use queries solely as local attention compressors to reduce spatial redundancy within images. In contrast, SurvQ introduces task-adaptive queries that automatically capture the most prognostically relevant information from both pathology and genomic data, while minimizing cross-modal redundancy and without relying on human-designed priors. This simple and lightweight formulation achieves strong performance and supports interpretable cross-modal aggregation without increasing architectural complexity.
>
> We believe this represents a practically meaningful and conceptually distinct contribution within the context of survival modeling, advancing multimodal learning in this domain. In the revised manuscript, we have added the QnA work to the Related Work section to better clarify these distinctions.
>
> **Q3: There is no ablation experiment on the number of queries.**
>
> A3: Thank you for this question. We did include an ablation on the number of queries in Supplementary Table 3, but we apologize if it was not sufficiently visible. For clarity, we reproduce the table here.
>
> To assess the effect of query numbers, we vary the query number for both modalities and evaluate performance. As shown in Table 1, the model benefits from a sufficient pool, about 300 queries for histology and 128 for genomics, while larger numbers offer diminishing returns. Notably, all configurations still surpass prior methods.
>
> Table 1: Ablation study on different numbers of learnable query of SurvQ
>
> | Hist. | Geno. | BRCA            | BLCA            | COADREAD         | HNSC            | STAD            | Avg.  |
> |-------|-------|-----------------|-----------------|------------------|-----------------|-----------------|-------|
> | 400   | 128   | 0.743 ± 0.046   | 0.657 ± 0.037   | 0.813 ± 0.095    | 0.669 ± 0.042   | 0.655 ± 0.032   | 0.708 |
> | 300   | 128   | 0.794 ± 0.062   | 0.677 ± 0.047   | 0.812 ± 0.079    | 0.653 ± 0.066   | 0.686 ± 0.051   | 0.724 |
> | 200   | 128   | 0.792 ± 0.045   | 0.652 ± 0.053   | 0.780 ± 0.099    | 0.655 ± 0.050   | 0.666 ± 0.025   | 0.709 |
> | 300   | 256   | 0.751 ± 0.082   | 0.673 ± 0.057   | 0.764 ± 0.119    | 0.677 ± 0.051   | 0.687 ± 0.036   | 0.710 |
> | 300   | 128   | 0.794 ± 0.062   | 0.677 ± 0.047   | 0.812 ± 0.079    | 0.653 ± 0.066   | 0.686 ± 0.051   | 0.724 |
> | 300   | 64    | 0.800 ± 0.039   | 0.679 ± 0.045   | 0.807 ± 0.098    | 0.649 ± 0.052   | 0.646 ± 0.010   | 0.716 |
>
> **Q4: Computational resource usage analysis.**
>
> A4:   Thank you for the great suggestion. We quantitatively evaluated the efficiency benefits of query compression (Table 2). SurvQ achieves approximately 40% faster inference and ~15% lower GPU memory usage compared to PIBD. This demonstrates that our ‘redundancy reduction’ mechanism provides tangible computational gains, not merely conceptual advantages. We have added this analysis to the Supplementary in the revised manuscript.
>
> Table 2: Ablation study on learnable vs. fixed queries
>
> | Method                 | BRCA            | COADREAD         |
> |------------------------|-----------------|------------------|
> | Fixed query            | 0.768 ± 0.057   | 0.778 ± 0.140    |
> | Ours (Learnable query) | 0.794 ± 0.062 | 0.812 ± 0.115 |

---

> > ### Author Response · Authors · 2025-11-24
> > **Response to Reviewer ubYy**
> >
> > **Q5: There are relatively few baselines for comparison, and most are not from 25 years ago.**
> >
> > A5: Thank you for the comment. We apologize for the confusion. In fact, we compared SurvQ against a broad set of competitive baselines, including recent multimodal methods such as MCAT (2021), Porpoise (2022), MOTCat (2023), SurvPath (2024), PIBD (2024), MMP (2024), and CCL (2025), which represent the state-of-the-art in multimodal survival prediction. These methods were published within the last few years (2021–2025), not 25 years ago.

---

> > > ### Comment · Reviewer_ubYy · 2025-11-25
> > > **Response**
> > >
> > > Dear Authors,
> > >
> > > Thank you for your response. It addressed some of my concerns, and as a result, I have increased the score to 4.
> > >
> > > Regarding Q4: I am looking for more concrete efficiency data, such as the specific FLops and memory usage relationships with metrics, as shown in Figure 3 (C) of Zhang S, Lin X, Zhang R, et al. "AdaMHF: Adaptive Multimodal Hierarchical Fusion for Survival Prediction," arXiv preprint arXiv:2503.21124, 2025. A textual description is not enough. Could you please provide a more detailed comparison like this, referencing similar studies and their data?
> > >
> > > Regarding Q5: Apologies for the misunderstanding in my previous comment. What I meant to express is that most of the baselines are not from recent years, particularly 2025, rather than 25 years ago. If you could include more up-to-date baselines from 2024 and 2025, it would help me to consider increasing the score further.
> > >
> > > Looking forward to your updated submission!

---

> > > > ### Author Response · Authors · 2025-11-26
> > > > **Response to Reviewer ubYy**
> > > >
> > > > We truly appreciate the reviewer replying to our response and considering raising the score.
> > > >
> > > > **Q1: Looking for more concrete analysis of computational resource usage.**
> > > >
> > > > A1:  Thank you for your question, and we apologize for misplacing the computational cost analysis table in our earlier response. We have included the inference time and GPU memory, which more accurately reflects practical computational cost, in the revised manuscript. For clarity, we reproduce the table here.
> > > >
> > > > Table 1: Ablation study on average inference time (ms) and GPU memory (GB)
> > > >
> > > > | Method | BRCA Inference Time (↓) | BRCA GPU (↓) | COADREAD Inference Time (↓) | COADREAD GPU (↓) |
> > > > |--------|---------------------------|---------------|-------------------------------|--------------------|
> > > > | PIBD   | 172                       | 1.42         | 170                           | 1.42              |
> > > > | SurvQ | 106                 | 1.21     | 106                       | 1.21          |
> > > >
> > > > **Q2: Include more up-to-date baselines from 2024 and 2025.**
> > > >
> > > > A2: Thank you for the suggestion. We have incorporated two additional recent baselines, MoME [1] (2024) and HSFSurv [2] (2025.9), into our main comparison table. Due to the large size of the full table, we present only the two newly added baselines below. The complete comparison can be found in Table 1 of the revised manuscript.
> > > >
> > > > | Model   | BRCA               | BLCA               | COADREAD           | HNSC               | STAD               | Avg.  |
> > > > |---------|--------------------|--------------------|--------------------|--------------------|--------------------|-------|
> > > > | HSFSurv | 0.771 ± 0.061      | 0.672 ± 0.031      | 0.761 ± 0.125      | 0.651 ± 0.042      | 0.674 ± 0.058      | 0.705 |
> > > > | MoME    | 0.768 ± 0.063      | 0.666 ± 0.022      | 0.785 ± 0.124      | 0.640 ± 0.054      | 0.673 ± 0.057      | 0.706 |
> > > > | **SurvQ** | **0.794 ± 0.062** | **0.677 ± 0.060** | **0.812 ± 0.115** | **0.653 ± 0.045** | **0.686 ± 0.053** | **0.724** |
> > > >
> > > > [1]  MoME: Mixture of Multimodal Experts for Cancer Survival Prediction, MICCAI 2024.
> > > >
> > > > [2]  HSFSurv: A hybrid supervision framework at individual and feature levels for multimodal cancer survival analysis, Medical Image Analysis 2025.

---

### Official Review · Reviewer_VhfN · 2025-10-25

**Soundness:** 3
**Presentation:** 3
**Contribution:** 2
**Rating:** 6
**Confidence:** 2

**Summary:**

This paper focuses on the problem of excessive redundancy in multimodal data in multimodal cancer survival analysis. Previous methods rely too much on prior knowledge, limiting the flexibility in capturing dynamic data changes and emerging patterns. This paper proposes SurvQ, conducting multimodal cancer survival analysis with learnable queries. By adaptively learning representative features in a data-driven manner, the method can reduce redundancy while preserving important information in multimodal data.

**Strengths:**

1. The proposed method is simple but effective. The approach generalizes the concept of query-based token compression (from DETR/BLIP-2) to the medical multimodal setting, replacing handcrafted prototype-based reductions with a data-driven mechanism.
2. This paper is well-written and easy to follow. The figures are clear.
3. The visualization of histology and genomic queries provides biological insight into the model’s internal representations.

**Weaknesses:**

1. While the empirical results are strong, the paper lacks deeper theoretical or information-theoretic analysis explaining why query-based bottlenecks improve generalization or reduce redundancy.
2. The evaluation is restricted to cancer survival prediction on TCGA datasets. It would strengthen the contribution to demonstrate the general applicability of SurvQ to other multimodal biomedical tasks.

**Questions:**

1. Are there any observed correlations between specific queries and known clinical or molecular subtypes?

---

> ### Author Response · Authors · 2025-11-24
> **Response to Reviewer VhfN**
>
> We thank the reviewer for the insightful comments.
>
> **Q1: While the empirical results are strong, the paper lacks deeper theoretical or information-theoretic analysis explaining why query-based bottlenecks improve generalization or reduce redundancy.**
>
> A1: Thank you for raising this question. Prior work by Set Transformer [1] provides a solid theoretical foundation for this design: they mathematically prove that a small set of learnable seed vectors (the same pattern as learnable queries) can universally approximate permutation-invariant set functions and effectively compress a large set into a compact, representative summary. This mechanism has since been validated empirically across many domains, including object detection (DETR) and multimodal representation learning (BLIP-2).
>
> Our design is also supported by well-established principles:
>
> (1) Bottleneck structures improve generalization by forcing the model to retain only task-relevant information
>
> (2) Token redundancy reduction is widely observed in WSI pathology, where most patches are background or low-informative. Query-based compression explicitly prevents overfitting to noisy or repetitive patches.
>
> Together, these insights show that using learnable queries to compact large sets is a well-known and principled mechanism.
> Our empirical results further confirm this intuition: SurvQ achieves higher survival accuracy with fewer tokens, significantly reduces inference time and GPU memory usage, and learns diverse, non-collapsed query representations. These findings collectively validate both the theoretical motivation and practical effectiveness of our approach
>
> [1] Set Transformer: A Framework for Attention-based Permutation-Invariant Neural Networks ICML2019.
>
> **Q2: The evaluation is restricted to cancer survival prediction on TCGA datasets.**
>
> A2:  Thank you for the suggestion. We agree that demonstrating broader applicability is valuable. In this work, we intentionally focus on TCGA cancer survival prediction, which is the standard benchmark used by nearly all recent multimodal pathology-genomics methods (e.g., SurvPath, PIBD). Evaluating on TCGA ensures direct comparability, clinical relevance, and consistency with prior literature.
>
> Importantly, SurvQ is not specialized to survival prediction. It introduces a general query-based multimodal bottleneck that is compatible with any paired image-omics or image-tabular task. The model architecture makes no assumption specific to TCGA, the same mechanism applies to multimodal prediction, regression, or representation learning.
>
> Extending SurvQ to additional multimodal biomedical tasks (e.g., drug response, molecular phenotype prediction, digital pathology and radiology) is a promising direction for future work.
>
> **Q3: Are there any observed correlations between specific queries and known clinical or molecular subtypes?**
>
> A3: Thank you for this question. Across different cancer types, several learned queries show biologically meaningful associations with tissue phenotypes that are characteristic of known molecular programs, as shown in Supplementary Figures 7 and 8. For example, in BLCA, queries enriched for lipid/steroid metabolism and NODAL/TGF-β signaling consistently highlight necrotic regions and epithelial-stromal transition zones, patterns commonly observed in basal/EMT-like tumors. In BRCA, queries related to TNF-α signaling and allograft rejection frequently activate in immune-infiltrated areas, which align well with the immune-hot and basal-like subtypes reported in prior studies.
>
> These examples illustrate that SurvQ’s learned queries capture biologically coherent cross-modal associations that reflect subtype-related histological patterns, even though no subtype supervision is used.

---

### Official Review · Reviewer_7uMb · 2025-10-30

**Soundness:** 2
**Presentation:** 3
**Contribution:** 1
**Rating:** 4
**Confidence:** 4

**Summary:**

This paper proposes SurvQ, a multimodal cancer survival analysis framework that integrates whole-slide histology images (WSIs) and transcriptomic profiles using learnable queries. Specifically, the authors introduce two sets of learnable query vectors that interact with unimodal features via cross-attention to extract representative pathology and genomic features, followed by a mixed self-attention module for multimodal fusion. The method is evaluated on five TCGA datasets and demonstrates better performance compared to prior multimodal and prototype-based methods.

**Strengths:**

1. Data-driven prototype learning via learnable queries provides a simple yet effective solution.
2. The model achieves better results across multiple TCGA cohorts, clearly outperforming both unimodal and prior multimodal baselines.
3. Visualization of attention maps and the top 6 pathways provides some interpretability, linking learned queries to meaningful histological and molecular patterns.

**Weaknesses:**

1. Conceptually incremental — mainly adapting the learnable-query idea from prior works (e.g., BLIP-2, DETR) to this setting.
2. The baseline design of the ablation study is weak; it doesn’t isolate the effect of “learnability.” A fixed/random query baseline would make the comparison fairer.
3. Interpretability focuses on unimodal patterns, cross-modal interactions (e.g., which histology regions relate to which genomic pathways) are not explored.
4. The number of queries is fixed for all datasets, which may not be optimal.

**Questions:**

1. Do the authors employ any mechanism (e.g., orthogonal regularization,  contrastive objectives) to encourage diversity or competition among the learnable queries? Otherwise, queries may collapse to similar representations.
2. Could the authors show cross-modal attention maps to verify the interaction between modalities?
3. Have the authors evaluated the computational cost or memory benefit of query compression? A quantitative comparison of efficiency would make the “redundancy reduction” claim more concrete.

---

> ### Author Response · Authors · 2025-11-24
> **Response to Reviewer 7uMb**
>
> We thank the reviewer for the insightful comments.
>
> **Q1: Conceptually incremental — mainly adapting the learnable-query idea from prior works (e.g., BLIP-2, DETR) to this setting.**
>
> A1:  Apologies for the misunderstanding. We would like to clarify that our contribution is not the introduction of the general learnable-query concept. This idea already exists and has been widely adopted across domains, including DETR, BLIP-2, and other areas such as image segmentation (e.g., Mask2Former [1] and SAM [2]).
>
> Instead, our key novelty lies in how we use learnable queries: SurvQ introduces task-adaptive queries that automatically capture the most prognostically relevant information from both histology and genomic data while minimizing redundancy, all without relying on human-designed priors. This simple and lightweight formulation enables strong performance and interpretable cross-modal aggregation without increasing architectural complexity. We believe this provides a practically meaningful and conceptually distinct contribution within the context of survival modeling, helping to advance multimodal learning in this field.
>
> [1] Masked-attention Mask Transformer for Universal Image Segmentation CVPR 2022
>
> [2] Segment Anything ICCV2023
>
> **Q2: The baseline design of the ablation study is weak; it doesn’t isolate the effect of “learnability.” A fixed/random query baseline would make the comparison fairer.**
>
> A2:  We thank the reviewer for this constructive suggestion. To directly assess the impact of query learnability, we additionally compare our learned queries against a fixed-query baseline, where the queries are randomly initialized and kept frozen (non-learnable) throughout training. As shown in Table 1, the learned queries substantially outperform the fixed-query baseline across all datasets. This evaluation has been added to the Supplementary.
>
> Table 1: Ablation study on fixed vs. learnable queries.
>
> | Method                 | BRCA            | COADREAD         |
> |------------------------|-----------------|------------------|
> | Fixed query            | 0.768 ± 0.057   | 0.778 ± 0.140    |
> | Ours (Learnable query) | 0.794 ± 0.062 | 0.812 ± 0.115 |
>
> **Q3: Interpretability focuses on unimodal patterns, cross-modal interactions (e.g., which histology regions relate to which genomic pathways) are not explored.**
>
> A3: Many thanks for these helpful comments. As suggested, we added an additional analysis to further explore the cross-modal interactions between histology and genomics. To illustrate these interactions, we randomly select one learnable genomic query and list its associated genomic pathways. We then use the multimodal mixed self-attention to identify the most related histology queries  and map them back to the original image patches to visualize the most representative regions.
>
> As shown in Supplementary Fig. 7 and 8, the learned genomic semantics (B) align well with the biologically coherent histology queries (C, D). For instance, in the BLCA case, the cross-modal associations identified by SurvQ reveal biologically interpretable correspondences between genomic pathways and histological patterns. Pathways related to lipid and steroid metabolism are predominantly linked to necrotic or lipid-rich tumor regions, which typically exhibit pale, loosely structured tissue morphology. In contrast, the NODAL/TGF-β signaling pathway corresponds to epithelial-stromal transition zones, reflecting regions where tumor cells interface with surrounding stromal components and where EMT-related biology is expected to occur. Additionally, pathways associated with purine salvage and telomere synthesis map to highly proliferative tumor areas characterized by densely packed, hyperchromatic nuclei.
>
> These observations collectively demonstrate that the learned cross-modal queries in SurvQ capture meaningful, biologically grounded interactions between histology and genomics, going beyond what standard unimodal models can explain.
>
> **Q4: The number of queries is fixed for all datasets, which may not be optimal.**
>
> A4: We follow the common practice in DETR-style architectures by using a fixed number of queries across all datasets to ensure fair comparison and reproducibility. Furthermore, as shown in Supplementary Table 3, we conducted an ablation study on the number of queries. Although the optimal choice varies slightly across datasets, our selected configuration provides consistently strong performance and offers a good balance between accuracy and stability. For these reasons, we adopt a unified setting for all datasets. We will explore adaptive query numbers in future work.

---

> > ### Author Response · Authors · 2025-11-24
> > **Response to Reviewer 7uMb**
> >
> > **Q5: Do the authors employ any mechanism (e.g., orthogonal regularization, contrastive objectives) to encourage diversity or competition among the learnable queries?**
> >
> > A5: No, we do not employ any explicit diversity-enhancing mechanisms. SurvQ naturally learns diverse and distinct compact representations. To verify this, we visualize the distributions of pathology and genomic queries in the t-SNE space (Figure 5) in the revised manuscript Supplementary,  which shows that they capture clearly differentiated feature patterns.
> >
> > **Q6: Have the authors evaluated the computational cost or memory benefit of query compression?**
> >
> > A6: Thank you for the great suggestion. We quantitatively evaluated the efficiency benefits of query compression (Table 2). SurvQ achieves approximately 40% faster inference and ~15% lower GPU memory usage compared to PIBD. This demonstrates that our ‘redundancy reduction’ mechanism provides tangible computational gains, not merely conceptual advantages. We have added this analysis to the Supplementary in the revised manuscript.
> >
> >  Table 2: Ablation study on average inference time (ms) and GPU memory (GB)
> >
> > | Method | BRCA Inference Time (↓) | BRCA GPU (↓) | COADREAD Inference Time (↓) | COADREAD GPU (↓) |
> > |--------|---------------------------|---------------|-------------------------------|--------------------|
> > | PIBD   | 172                       | 1.42         | 170                           | 1.42              |
> > | SurvQ | 106                 | 1.21     | 106                       | 1.21          |

---

### Official Review · Reviewer_sERY · 2025-11-01

**Soundness:** 2
**Presentation:** 3
**Contribution:** 2
**Rating:** 2
**Confidence:** 5

**Summary:**

This work proposes to utilize two sets of learnable queries to extract representative features via cross-attention, while multimodal mixed self-attention is leveraged to model cross-modal interactions.

**Strengths:**

1. The motivation is clear, and the manuscript is well-written.
2. The performance of the proposed method is superior to SOTA approaches.
3. The effectiveness of each component is validated by ablation studies.

**Weaknesses:**

1. The novelty is incremental and very simple. The core idea follows PIBD by using a set of learnable parameters to capture representative features for each modality. The difference is that PIBD enforces a risk level constraint to assist the model in learning discriminative features, while there is no prior constraint in the proposed method. Additionally, the idea of learnable queries has been explored in G-HANet [1], which has validated its effectiveness.
2. The insight about why it works is lacking. Given that there is no explicit constraint for the modelling, although the story is well-told, I'm still confused about why it achieves the intended purpose without any explicit guidance or constraint, and why it is better than PIBD with explicit conditions.

[1] Wang Z, Zhang Y, Xu Y, et al. Histo-genomic knowledge association for cancer prognosis from histopathology whole slide images[J]. IEEE Transactions on Medical Imaging, 2025.

**Questions:**

See weakness.

---

> ### Author Response · Authors · 2025-11-24
> **Response to Reviewer sERY**
>
> We thank the reviewer for the insightful comments.
>
> **Q1: The novelty is incremental and very simple.**
>
> A1:  In terms of the fundamental model formulation, we agree that our approach follows the general design philosophy of SurvPath [1], but adopts a more simplified and compact representation and processing pipeline. This trend is also shared by several recent works, including MMP [2], PIBD [3], and CCL [4], each aiming to compress massive information into more compact and informative representations compared with their original baselines. For instance, MMP leverages morphological grouping to cluster large numbers of patches and applies Optimal Transport for cross-modal alignment. PIBD incorporates survival-related constraints (e.g., risk levels or censorship patterns) at the bag level to enforce more discriminative cluster features.
>
> Although these approaches successfully reduce redundancy, they are heavily shaped by their human-designed priors. While such priors can be useful, they may also restrict model flexibility. For example, MMP lacks risk-aware supervision, and PIBD may fail to capture morphology-driven structures unless explicitly provided.
>
> In contrast, our work introduces a fully data-driven formulation based on learnable queries, without relying on hand-crafted priors or complex architectural assumptions. While conceptually simple, SurvQ provides a flexible formulation for multimodal survival prediction that enables compact representation learning and full cross-modal alignment, while also offering faster inference and superior performance.
>
> [1] Modeling dense multimodal interactions between biological pathways and histology for survival prediction. CVPR2024
>
> [2] Multimodal Prototyping for cancer survival prediction ICML2024
>
> [3] Prototypical Information Bottlenecking and Disentangling for Multimodal Cancer Survival Prediction ICLR2024
>
> [4]  Cohort-Individual Cooperative Learning for Multimodal Cancer Survival Analysis TMI2025
>
> **Q2:  The idea of learnable queries has been explored in G-HANet, which has validated its effectiveness.**
>
> A2:  Thank you for pointing out G-HANet, which also employs learnable queries. However, the motivation and usage of learnable queries in G-HANet differ substantially from those in SurvQ. G-HANet utilizes learnable queries to capture histo-genomic associations through genome reconstruction during training. At inference time, these learned queries serve as surrogates for genomic information, allowing the model to operate without genomic inputs. In contrast, SurvQ employs learnable queries for a fundamentally different purpose: to compactly encode massive multimodal information without relying on complex human-designed priors. By comparison, G-HANet still operates on bag-level features to achieve its objective. We have added a detailed comparison with G-HANet in the Related Work section of the revised manuscript to clarify these distinctions.
>
> **Q3:  The insight about why it works is lacking.**
>
> A3:  We summarize the key insight here: the fundamental idea behind reducing redundancy is to cluster large, high-dimensional feature sets into compact and representative embeddings under certain constraints. Existing methods such as PIBD, CCL, and MMP all follow this philosophy and rely on human-designed priors to guide the clustering process. PIBD and CCL impose survival-related constraints (e.g., risk levels or censorship patterns) to group patients into predefined risk bands, while MMP uses morphology-derived grouping to structure the feature space.
>
> However, relying on a single type of constraint does not guarantee that the resulting clusters are the most informative or representative for the model. These constraints may even be complementary rather than mutually sufficient: for example, MMP lacks risk-aware supervision, whereas PIBD may fail to capture morphology-driven structure unless it is explicitly encoded.
>
> In contrast, we do not provide any specific or explicit human-designed guidance or constraints. Our method learns compact representations entirely through data-driven learnable queries, using only the survival prediction objective to supervise the learning process without relying on handcrafted priors. This paradigm has already proven effective in other domains, such as DETR and BLIP-2. Remarkably, the model is able to autonomously discover meaningful structure. As shown in Figures 4-8 in Supplementary, SurvQ learns distinct, diverse, and interpretable compact representations, which ultimately lead to improved performance.

---

> > ### Comment · Reviewer_sERY · 2025-11-26
> >
> > Thank you for the detailed responses. I appreciate the explanations provided, though I feel that some of my initial concerns may not have been fully addressed yet.
> >
> > Regarding the use of prior knowledge, I understand it can be a double-edged sword. The authors mention that avoiding manual prior constraints is an advantage of their approach. However, I was wondering how the model ensures a compact latent space without any explicit constraints. It would be very helpful if the authors could provide empirical evidence or a theoretical analysis to demonstrate the model's ability to learn compact representations effectively.
> >
> > Additionally, you noted that "this paradigm has already proven effective in other domains, such as DETR and BLIP-2." Could you kindly clarify the specific contribution of your work in this context?
> >
> > Thank you for considering my questions.

---

> > > ### Author Response · Authors · 2025-11-26
> > > **Response to Reviewer sERY**
> > >
> > > Thank you for your follow-up questions.
> > >
> > > **Q1: Regarding the use of prior knowledge, I understand it can be a double-edged sword. The authors mention that avoiding manual prior constraints is an advantage of their approach. However, I was wondering how the model ensures a compact latent space without any explicit constraints. It would be very helpful if the authors could provide empirical evidence or a theoretical analysis to demonstrate the model's ability to learn compact representations effectively.**
> > >
> > > A1:  We appreciate that we have a shared understanding that manually imposed prior constraints may limit the flexibility of survival prediction models. Our method is specifically designed to avoid such hand-crafted priors by introducing learnable queries, enabling the model to discover compact and informative representations without relying on pre-defined structures. We have provided extensive quantitative and qualitative evidence to support this design, including state-of-the-art performance comparisons, ablation studies demonstrating the effectiveness of learnable queries, interpretability analyses of intra- and inter-modal interactions, and visualizations of the learned query distributions. We apologize if the explanation in the main paper was not sufficiently clear or systematic. For clarity, we summarize the step-by-step evidence below:
> > >
> > > 1. **Overall performance**: We begin with a comprehensive comparison against state-of-the-art methods (**Table 1**), demonstrating that SurvQ achieves strong performance across multiple benchmarks.
> > >
> > > 2. **Effectiveness of learnable queries**: We then provide ablations (**Table 2 and Table 4**) showing that learnable queries play a crucial and effective role in performance gains.
> > >
> > > 3. **Interpretability**: Beyond numerical results, we offer clear visual evidence (**Figures 4, 6, 7, and 8**) that SurvQ learns interpretable and compact representations within and across modalities.
> > >
> > > 4. **Non-collapsing query representations**: Finally, to verify that the learnable queries do not collapse to similar features, we include a t-SNE visualization (**Figure 5**) showing diverse and well-separated query embeddings.
> > >
> > > We believe these quantitative and qualitative results together provide strong evidence for the effectiveness and interpretability of SurvQ.
> > >
> > > **Q2:  Additionally, you noted that "this paradigm has already proven effective in other domains, such as DETR and BLIP-2." Could you kindly clarify the specific contribution of your work in this context?**
> > >
> > > A2: We would like to clarify that our contribution is not the introduction of the general learnable-query concept. This idea already exists and has been widely adopted across domains, including DETR, BLIP-2, and other areas such as image segmentation (e.g., Mask2Former [1] and SAM [2]).
> > >
> > > **Our contribution lies in how learnable queries are used in the context of survival prediction**, which differs fundamentally from their role in detection/segmentation or vision-language models. Specifically, SurvQ introduces task-adaptive survival queries that automatically extract the most prognostically relevant information from both histology and genomic data while avoiding hand-crafted cross-modal priors. These queries are optimized end-to-end to (1) capture compact multimodal representations, (2) reduce feature redundancy, and (3) enable interpretable aggregation across modalities.
> > >
> > > This formulation is simple, lightweight, and requires no additional architectural complexity, yet it yields state-of-the-art performance and produces meaningful intra- and inter-modal interactions. We believe this offers a practically useful and conceptually distinct contribution to survival modeling, advancing multimodal learning in this domain.
> > >
> > > [1] Masked-attention Mask Transformer for Universal Image Segmentation, CVPR 2022.
> > >
> > > [2] Segment Anything, ICCV 2023.

---

> > > > ### Comment · Reviewer_sERY · 2025-11-27
> > > >
> > > > I may not have been clear in my previous question, and I apologize for any confusion. I completely understand that your method is empirically effective. My question is, without explicit constraints, how do you prove the learned representation space is compact via learnable queries? How much redundancy is reduced? Besides performance gains, the direct evidence that proves the compactness should be presented.

---

> > > > ### Comment · Reviewer_sERY · 2025-11-27
> > > >
> > > > Regarding the contribution on "how learnable queries are used in the context of survival prediction", I may be missing something, but their implementation appears to be a well-established technique. Could you elaborate on their unique adaptation for survival prediction?

---

> > > > > ### Author Response · Authors · 2025-11-28
> > > > > **Response to Reviewer sERY**
> > > > >
> > > > > Thanks for your quick response and follow-up questions.
> > > > >
> > > > > **Q1: I may not have been clear in my previous question, and I apologize for any confusion. I completely understand that your method is empirically effective. My question is, without explicit constraints, how do you prove the learned representation space is compact via learnable queries? How much redundancy is reduced? Besides performance gains, the direct evidence that proves the compactness should be presented.**
> > > > >
> > > > > A1:  We are glad that we share the understanding that the proposed method is empirically effective.
> > > > > To clarify why the learned representation is compact, our learnable queries preserve only the essential, task-relevant information from the original input while using far fewer tokens.
> > > > >
> > > > > To assess how much information is preserved by the learnable queries, we perform **PCA** on the original histology and omics features and compute the variance explained by their principal subspace. We then project the learned queries onto this PCA basis and measure the proportion of variance retained within the shared subspace. As shown in Table 1, the queries retain **94.6%** of the histology variance and **96.7%** of the omics variance captured by the original representations. The difference between 99.05% and 94.55% reflects the fact that the raw patch features contain substantial high-frequency appearance variations (e.g., staining variability, morphological noise, tile-level artifacts) that contribute to the overall variance but are largely irrelevant to survival prediction. The learned queries deliberately suppress these noisy or task-irrelevant directions during end-to-end learning, focusing instead on the lower-dimensional, task-relevant subspace. For omics features, the variance gap between the original representation (99.04%) and the learned queries (96.7%) is smaller than in histology. This is expected because omics embeddings are already low-noise, highly structured, and free from spatial or staining artifacts. These results demonstrate that SurvQ achieves effective information-preserving compression: the learned queries substantially reduce the token count while maintaining nearly all major modes of variation present in the original features.
> > > > > At the same time, the queries remain diverse and well-separated (Figure 5), demonstrating that the model does not collapse into similar or redundant tokens.
> > > > >
> > > > > Furthermore, Figures 4, 6, 7, and 8 provide visual evidence that SurvQ learns **interpretable** and compact **cross-modal representations** organized purely under survival supervision.
> > > > >
> > > > > Regarding redundancy reduction, SurvQ reduces histology tokens from over 10,000 to 300 and pathways tokens from 331 to 128, achieving substantial compression while still improving performance. This reduction also enables full self-attention across modalities, enhancing both predictive performance and interpretability. Compared to representative prototype-based baselines that also focus on redundancy reduction, SurvQ achieves **~40% faster inference** and **~15% lower GPU memory usage** than PIBD, showing that our compact representation brings concrete computational benefits rather than only empirical gains.
> > > > >
> > > > > We have added the analysis to the Supplementary in the revised manuscript. We hope this addresses your concerns, and we are happy to clarify further if needed.
> > > > >
> > > > > Table 1: PCA-based information preservation analysis on COADREAD dataset.
> > > > >
> > > > > | Modality   | Var(Original features) | Var(Learned queries) |
> > > > > |------------|-------------------------|------------------------|
> > > > > | Histology  | 99.05%                  | 94.55%                 |
> > > > > | Omics      | 99.04%                  | 96.70%                 |
> > > > >
> > > > > Table 2: Ablation study on average inference time (ms) and GPU memory (GB).
> > > > >
> > > > > | Method | BRCA Inference Time (↓) | BRCA GPU (↓) | COADREAD Inference Time (↓) | COADREAD GPU (↓) |
> > > > > |--------|---------------------------|---------------|-------------------------------|--------------------|
> > > > > | PIBD   | 172                       | 1.42         | 170                           | 1.42              |
> > > > > | SurvQ | 106                 | 1.21     | 106                       | 1.21          |

---

> > > > > > ### Author Response · Authors · 2025-11-28
> > > > > > **Response to Reviewer sERY**
> > > > > >
> > > > > > **Q2: Regarding the contribution on "how learnable queries are used in the context of survival prediction", I may be missing something, but their implementation appears to be a well-established technique. Could you elaborate on their unique adaptation for survival prediction?**
> > > > > >
> > > > > > A2: Thank you for your question. Our contribution is **not** introducing learnable queries or attention, which are well established. Instead, our novelty lies in how we **adapt** them to address a **core limitation** in survival analysis.
> > > > > >
> > > > > > (1) **Removing reliance on hand-crafted priors.**
> > > > > > Current survival models depend heavily on human-defined priors. SurvQ instead uses learnable queries to automatically extract compact, task-relevant histology and genomic representations optimized directly for survival prediction, fundamentally different from how queries are used in other domains.
> > > > > >
> > > > > > (2)  **A simple and effective cross-modal interaction design.**
> > > > > > With these compact representations, our simple mixed-attention mechanism becomes highly effective, as it directly benefits from the learnable-query design without adding architectural complexity.
> > > > > >
> > > > > > Our pipeline is intentionally simple, **without introducing additional constraints or unnecessary architectural complexity**, while still achieving **stronger performance and interpretability**. We hope this clarifies the novelty of our approach.

---

### Author Response · Authors · 2025-12-03
**Global Response: Core Contributions of our Paper and Rebuttal Summary for the new AC**

**(IV) Current Status**

- Reviewer **@sERY** (**Rating: 2**): We appreciate the constructive discussion with reviewer **@sERY**. Through the exchange, we **reached agreement** that prior knowledge “*can be a double-edged sword*” and that our model is “*empirically effective*.” The remaining concern is whether “***the learned representation space is compact via learnable queries***.”  We have now provided additional evidence through PCA-based analysis and redundancy-reduction measurements to support this claim.

- Reviewer **@ubYy** (**Rating: 2 -> 4, 25 Nov**): We partially addressed the concerns regarding novelty, fixed query number. The reviewer **@ubYy** has increased the rating from 2 to 4, and **indicated to further increase the rating** if a computational usage table and more up-to-date baselines from 2024 and 2025 are included. We have carefully addressed both points in the revised manuscript.

- We believe our additional experiments and revisions effectively resolve the open questions raised by **@VhfN** (**Rating 6**) and **@7uMb** (**Rating 4**).

We hope this summary aids your final decision. Thanks for your precious time!

Best regards,

The Authors

---

### Author Response · Authors · 2025-12-03
**Global Response: Core Contributions of our Paper and Rebuttal Summary for the new AC**

**Dear PCs, SACs, ACs, and Reviewers,**

Thank you for handling our submission. We understand this is a busy period, so we provide a concise summary of our rebuttal updates to assist in your final assessment—particularly as Reviewers **@7uMb** and **@VhfN** have not yet responded to our revisions.

**(I) Core Contribution**

We propose **SurvQ**, which introduces **task-adaptive queries** that automatically capture the most prognostically relevant information from both histology and genomic data while minimizing redundancy, all **without relying on human-designed priors**. This simple and lightweight formulation enables strong performance and interpretable cross-modal aggregation **without introducing additional constraints or unnecessary architectural complexity**.

**(II) Reviewer Consensus on Strengths**

The reviewers consistently highlight four key strengths:

**1. Architectural Elegance and Efficiency:**

Reviewers **@7uMb**, **@VhfN**, and **@ubYy** consistently highlighted that our method “*provides a simple yet effective solution*”.

**2. Superior Empirical Performance:**

Reviewers **@sERY**, **@7uMb**, and **@ubYy** agreed that “*the performance of the proposed method is superior to SOTA approaches*”, and that “*the effectiveness of each component is validated by ablation studies*”.

**3. Strong Interpretability:**

Reviewers **@7uMb**, **@VhfN**, and **@ubYy** appreciated that “*the visualization of histology and genomic queries provides biological insight into the model’s internal representations*”.

**4. Clarity and Quality of Writing:**

Reviewer **@sERY** noted that “*the motivation is clear, and the manuscript is well-written*”, which was echoed by **@VhfN**, who remarked that “*this paper is well-written and easy to follow*”.

**(III) Key Revisions & New Experiments**

We utilize the discussion period to conduct targeted experiments addressing the specific concerns of all reviewers, including those who remain silent.

**1. Novelty (Addressing @sERY, @7uMb, @ubYy)**

- We declare that our contribution **is not** the introduction of the general learnable-query concept, which has been explored in other domains, including DETR, BLIP-2, Mask2Former, and SAM.

- Instead, our **key novelty** lies in how we use learnable queries within the context of survival modeling, advancing multimodal learning in this domain.

- Our simple and lightweight formulation achieves **strong performance** and supports **interpretable** cross-modal aggregation **without increasing architectural complexity**.

**2. Insight (Addressing @sERY)**

- The reviewer asked why our method could work and could be better than PIBD with explicit conditions. We provide evidence from several angles.

- First, as noted by **@sERY**, “*the performance of the proposed method is superior to SOTA approaches*”, and "*the effectiveness of each component is validated by ablation studies*". We further show that the **learned queries do not collapse** (Figure. 5) and provide **interpretable visualizations** of both **unimodal and cross-modal** queries (Figure. 4, 6–8).

- SurvQ **significantly reduces token counts**, 10k -> 300 for histology and 331 -> 128 for omics, yet preserves essential information. To demonstrate this compactness, we perform **PCA analysis** on the original features and measure how much of their variance is retained by the learned queries within the same principal subspace. As shown in Table 6, the queries **preserve nearly all** the meaningful variance while **suppressing noisy or task-irrelevant** directions, yielding a more focused and discriminative representation.

- Finally, Table 5 shows that this **reduction in redundancy** translates into clear computational benefits.

**3. Cross-modal interactions visualization (Addressing @7uMb, @VhfN)**

- We have presented the visualization of cross-modal interactions in Figure. 7–8.

**4. Computational resource usage (Addressing @7uMb, @ubYy)**

- We have compared inference time and GPU usage with PIBD on two datasets, as reported in Table 5.

**5. Fixed query numbers (Addressing @7uMb, @ubYy)**

- We follow the common practice in DETR-style architectures by using a fixed number of queries across all datasets to ensure fair comparison and reproducibility. We conducted an ablation study on the number of queries (Table 3).

**6. Query learnability ablation (Addressing @7uMb)**

- We have provided the ablation in Table 4.

**7. Mechanism to encourage diversity (Addressing @7uMb)**

- We have visualized the distributions of pathology and genomic queries in the t-SNE space (Figure 5), showing that they capture clearly differentiated feature patterns.

---

### Meta-Review · Area_Chair_rcmC · 2026-01-01

**Summary:**

Reviewers generally found SurvQ well written and empirically strong on five TCGA cohorts, with ablations and qualitative visualizations suggesting meaningful unimodal and some cross-modal interpretability. The main concerns were that the core idea—query-based token compression—is conceptually incremental relative to prior prototype/bottleneck and learnable-query methods (e.g., PIBD and query-based attention work), and that the paper initially lacked direct evidence quantifying “redundancy reduction/compactness” beyond performance. Several reviewers also requested stronger experimental substantiation (e.g., fixed/random query baselines, ablations on query count, clearer cross-modal attention visualizations, and concrete efficiency reporting such as runtime/memory/FLOPs and more up-to-date baselines). Overall, this paper is below the bar of acceptance.

**Reviewer Concerns:**

Concerns regarding specific framework and hyperparameter design choices were largely addressed during the rebuttal through additional ablations and experimental analyses. However, the more fundamental questions about technical novelty and a clear explanation of why the proposed framework is intrinsically effective remain insufficiently addressed.

**Reviewer Scores:**

Reviewer VhfN would maintain a score of 6, albeit with low confidence (2). Reviewer ubYy increased their rating from 2 to 4. In contrast, Reviewer sERY and Reviewer 7uMb are unlikely to raise their scores (and may remain negative), as their primary concern—limited technical novelty—does not appear to be fully addressed.

---

### Decision · Program_Chairs · 2026-01-26

Reject